

# Once bitten, twice shy: experienced regret and non-adaptive choice switching

Francesco Marcatto, Anna Cosulich and Donatella Ferrante

Department of Life Sciences, University of Trieste, Italy

## ABSTRACT

When a good decision leads to a bad outcome, the experience of regret can bias subsequent choices: people are less likely to select the regret-producing alternative a second time, even when it is still objectively the best alternative (non-adaptive choice switching). The first study presented herein showed that nearly half of participants experiencing regret rejected a previous alternative they had recognized as the best one, and chose a non-optimal alternative instead. The second study investigated the mechanism underlying this bias, and results supported the hypothesis that this non-adaptive choice switching is caused by inhibition of the previous decision (direct effect of experienced regret), rather than by increased sensitivity to anticipated regret in subsequent choices (indirect effect of experienced regret mediated by anticipated regret).

## INTRODUCTION

Regret is an emotion with strong cognitive roots, based on a comparison between "what is" and "what might have been," that is, between the outcome we actually obtained and a better outcome we would have obtained, had we chosen differently (*Van Dijk & Zeelenberg, 2005*). Regret happens: it is unavoidable. Indeed, how many of us have not wanted to just kick ourselves, at least once, for something we did (or did not do)? When we look back and see that things would have turned out better had we made a different choice, we frequently experience this unpleasant emotion (*Marcatto & Ferrante, 2012*). In order to feel regret, we do not need to actually observe that we could have obtained a better outcome; it is sufficient to imagine it (*Kahneman, 1995*; *Bar-Hillel & Neter, 1996*). In decision-making research, regret is usually differentiated from disappointment: Both are negative emotions that arise from counterfactual comparison, but while regret results from a comparison between an actual outcome and a better outcome that might have occurred had we made a different choice, disappointment stems from comparing an obtained outcome with a better outcome that might have occurred had we made the same choice. This distinction is particularly relevant because these emotions have distinguishable behavioral consequences for decision making (see, e.g., *Van Dijk & Zeelenberg, 2002*; *Zeelenberg et al., 2000*).

Why do we feel regret? Isn't it useless to cry over spilled milk? It is generally assumed that although painful, regret is functional because it helps us make better decisions. According to the Regret Theory (*Bell, 1982*; *Loomes & Sugden, 1982*), we can anticipate

Corresponding author
Francesco Marcatto,
fmarcatto@units.it

the possible emotional consequences of our choices before we make them, and use these anticipated emotions to guide our choices (*Janis & Mann, 1977*; *Mellers, Schwartz & Ritov, 1999*). Thus, fear of possible future regret let us usually avoid risky behaviors and painful experiences (e..g., *Zeelenberg, 1999a*; *Wright & Ayton, 2005*). Moreover, experienced regret (or retrospective regret) may help us to prevent similar mistakes in the future, since it makes the mistakes more painful (*Zeelenberg, 1999b*; *Saffrey, Summerville & Roese, 2008*). According to a popular quote, "*Regret is insight that comes a day too late.*" In fact, when we think that we should have behaved differently, we learn something, for example that we should avoid the restaurant that served bad food or that drinking too much has its downsides. Thus, being able to look back on and evaluate our past choices allows us to modify future behavior and presumably make better decisions.

## Regret and adaptive choices

The idea of an adaptive role of the cognitive-based emotion of regret is supported by many research findings. Many studies have shown that people are regret averse: they spontaneously anticipate future regret and behave in ways that minimize the possibility of experiencing this negative emotion, for example by avoiding risky choices (*Richard, Van der Pligt & de Vries, 1996*; *Wright & Ayton, 2005*), or by avoiding feedback about foregone alternatives (*Larrick & Boles, 1995*; *Ritov, 1996*; *Zeelenberg et al., 1996*; *Hoelzl & Loewenstein, 2005*; *Van de Ven & Zeelenberg, 2011*).

Neuropsychological studies show that patients with selective lesions to the orbitofrontal cortex, an area associated with the processing of counterfactual comparisons, perform worse than controls in repeated gambling tasks, since they cannot learn from their prior emotional experiences (*Camille et al., 2004*; *Coricelli, Dolan & Sirigu, 2007*).

Recently, *O'Connor, McCormack & Feeney (2014)* investigated the development of regret in children, and found that when children begin to experience regret (between the ages of 5 and 7 years), the quality of their subsequent decision making improves. The development of regret allows children to learn from their previous choices and thus to make better choices when faced with the same situation again, a behavior that the authors have termed *adaptive choice switching*.

A similar regret-based learning process, called regret matching, has also been implemented in algorithms used in game theory. According to this procedure, in repeated games the player changes his current strategy for a foregone alternative that would have given a higher payoff in the past (*Hart & Mas-Colell, 2000*; *Hart, 2005*; *Marchiori & Warglien, 2008*). Regret matching is a simple adaptive procedure, based only on comparison of the realized payoff and the foregone payoffs, leading in the long run to a sophisticated solution of the game (correlated equilibrium) (*Coricelli, Dolan & Sirigu, 2007*).

## Non-adaptive choice switching

Is regret-induced choice switching always adaptive? What happens, for example, when people experience regret following an optimal decision, and are faced with the same choice again? When regret is the unfortunate consequence of a good decision, or when previous

outcomes are not related to subsequent choices, switching choices blindly could lead to biased decisions.

Do people actually make "bad decisions" as a consequence of regret? Some literature findings suggest that this can happen. Preliminary evidence was observed in a study conducted by *Zeelenberg & Beattie (1997)*. In one of their experiments, participants played two rounds of Ultimatum Game as offerers. After the first round, they were told that their offer was either 2 Guilders or 10 Guilders higher than the responder's minimal acceptable offer. Participants discovering they had offered 10 Guilders too much experienced regret and lowered their offer in the subsequent round thereby, even if they knew that they were playing with a new responder, whose minimal acceptable offer could have easily differed from the previous responder's offer. Similar results were found also in neuroimaging studies conducted by Büchel, Brassen and colleagues (*Büchel et al., 2011*; *Brassen et al., 2012*) aimed at investigate the neural mechanisms for how missed opportunities influence future choices. Their results showed that, in a sequential risk taking task, following a missed opportunity (e.g., you would have obtained a higher gain, if you had risked more) participants increased the risk taken in the next round, despite the fact that consecutive rounds were explicitly independent, and this behavior was paralleled by signal changes in the ventral striatum, a neural structure involved in regret processing.

*Ratner & Herbst (2005)* found another evidence for this effect. Their experiments were based on a scenario methodology requiring participants to choose between two brokers, one of whom was described as having a better success rate than the other. Obviously, most participants chose the broker with the better success rate. Half of them were then informed that the selected broker had succeeded, the others were told that their broker had failed, and all of them were then asked to imagine which of the two brokers they would choose the next time. Results showed that participants whose broker had failed regretted their decision and reported a lower intention to select the same broker again, even if she still had the best success rate.

## Choice switching: experienced vs. anticipated regret

Since most studies investigating the behavioral consequences of regret have adopted repeated gambling tasks involving many trials, it is usually difficult to disentangle the effect of anticipated from experienced regret. Indeed, *O'Connor, McCormack & Feeney (2014)* hypothesized two mechanisms that might explain the choice switching behavior. The first one is a simple process, consisting in remembering that a particular option yielded a poor result in the past, thus avoiding it when faced with the same choice again (*direct effect of experienced regret*, henceforth *direct effect*). According to the alternative explanation, the effect of experienced regret on subsequent choices is mediated by anticipated regret: A recent experience of regret could prime the anticipation of regret in the following choices, leading to increased regret aversion (*indirect effect of experienced regret mediated by anticipated regret*, henceforth *indirect effect*). The finding that the same neural circuitry mediates the experience of regret and its anticipation, emerged in brain imaging studies (*Coricelli et al., 2005*), provide support for the latter hypothesis. On the other hand, in the

*O'Connor, McCormack & Feeney (2014)* studies on regret in children provided evidence for the direct effect mechanism: The authors found a choice switching behavior in children who have already developed the ability to experience regret, but were still incapable of anticipating it, thus showing that experienced regret can affect a future choice independent of anticipated regret.

## The current studies

On the basis of the aforementioned considerations, the present studies were aimed at (i) providing further evidence of a non-adaptive choice switching behavior using real choices (Study 1), and at (ii) trying to shed light in the process that underpin this bias, by using a task in which the two hypothesized mechanisms (direct effect vs. indirect effect) would lead to different behaviors (Study 2).

Both studies adopted the choice switching paradigm (*O'Connor, McCormack & Feeney, 2014*). Specifically, a first choice was followed by feedback about the respective outcomes of the chosen and non-chosen alternatives. The aim was to induce either regret (i.e., you obtained a bad outcome, and the non-chosen alternative would have been better) or disappointment (i.e., you obtained a bad outcome and even the non-chosen alternative would have produced the same outcome).[1] Afterwards, a second choice was presented, either similar (Study 1) or different (Study 2) from the previous. Unlike previous studies on experienced regret, this experimental design made it possible to disentangle the specific effect of regret from a more general effect caused by disappointment for the negative outcome. It was hypothesized that, if regret influences subsequent decisions, participants who receive the regret feedback should switch from a good decision to a non-optimal one in a subsequent choice, as opposed to participants receiving the disappointment feedback.

## STUDY 1

### Method

#### Participants and design

A total of 105 students (70 women and 35 men; mean age = 22.2 years, $SD = 3.25$) from Trieste University voluntarily participated in the study and were randomly assigned to either the regret or disappointment condition. Participants were tested individually and were advised about the data retained and that anonymity was fully ensured, no sensitive data were collected.

#### Procedure

Participants played two rounds of a simplified computer-version of Blackjack against the computer-played dealer. To increase overall motivation and involvement in the game, each round had a 5-Euro prize. Before beginning the first round, participants received instructions on Blackjack rules. Participants started playing by turning their initial two cards face up, and then had to choose whether to take more cards or to stop, by pressing two buttons labeled "draw" and "stop," respectively. Both dealer's initial cards were face down, to prevent participants from using casino Blackjack strategies like basic (for a review on the psychology of playing Blackjack, see *Keren & Wagenaar, 1985*). The game

---

[1] Disappointment is often induced by showing the bad outcome only; participants do not actually see what the counterfactual outcome would have been (e.g., *Mellers, Schwartz & Ritov, 1999*). We decided to also give participants a feedback in disappointment conditions, to avoid the possibility that participants spontaneously produced regret-inducing counterfactual thoughts (e.g., "if only I had chosen differently, I would have obtained a better outcome").

## PeerJ

[2] Participants choosing to take another card at "18" were not assigned to the experimental conditions and did not play the second round.

was rigged so that in the first round all participants received the same cards (a "2," a "6" and a picture card), thus achieving the score of 18. We assumed that participants would have recognized that stopping was the optimal decision after having achieved the score of 18. In any event, participants who decided to draw another card would receive an "8," losing the game thereby. The dealer played after the participants and won by scoring 19 (with a picture card and a "9"); all participants therefore lost. Participants who decided to stop at "18" then received a feedback about what would had happened, had they chose to take another card. [2] Participants in the regret condition were shown that they would have drawn a "2," winning thereby with a score of 20 (regret feedback), while participants in the disappointment condition were shown that they would have drawn an "8," and that they therefore would have lost anyway (disappointment feedback). Participants' emotional reactions were assessed with the Regret and Disappointment Scale (*Marcatto & Ferrante, 2008*), a six items scale measuring regret, disappointment and the intensity of negative affect resulting from the bad outcome.

In the second round, participants achieved the score of 18 again, but this time with different cards (an "8" and a picture card), and they once more had to choose to either take another card or to stop. We therefore expected that, if previously experienced regret can induce non-adaptive choice switching, participants in the regret condition would less likely choose the optimal alternative ("to stop") than the participants in the disappointment condition would.

## Results

In the first round of Blackjack, most participants (90 out of 105, 86%) decided to stop after scoring 18. As hypothesized, participants correctly recognized that stopping was the optimal decision. The data from the 15 participants who decided to take another card were excluded from the analyses reported in the next section.

### Emotional reaction ratings

As reported in Table 1, the two conditions induced similar levels of negative affect ($t = .20$, $df = 88$, $p = .84$, $d = 0.04$), but different types of specific emotions: The regret score was higher in the regret condition than in the disappointment condition ($t = 7.80$, $df = 88$, $p < .001$, $d = 1.65$), and the disappointment score was higher in the disappointment condition than in the regret condition ($t = 4.98$, $df = 88$, $p < .001$, $d = 1.05$). Due to the key role of chance in Blackjack, high disappointment scores were expected for both conditions.

### *Second Blackjack round choices*

In the second round, 44% (20 out of 45) of the regret condition participants decided to take another card after having scored 18, whereas only 24% (11 out 45) of the disappointment condition participants decided to do so ($\chi^2_{(1,90)} = 3.99$, $p < .05$, Cramér's $V = .210$).

A logistic regression was then conducted to assess whether the participants' choices in the second round could be significantly predicted by the following variables: Type of feedback (regret or disappointment), negative affect, regret score and disappointment score. Since predictors were expected to be highly correlated, a stepwise method with

Table 1 **Mean emotional reaction ratings in the two experimental conditions.** Ratings ranged from 1 to 7.

| | Regret condition | | Disappointment condition | |
|---|---|---|---|---|
| | Mean | SD | Mean | SD |
| Negative affect | 3.16 | 1.55 | 3.09 | 1.58 |
| Regret score | 3.21 | 1.50 | 1.31 | 0.63 |
| Disappointment score | 3.68 | 1.45 | 5.23 | 1.50 |

[3] Other predictors excluded from the final regression model: Type of feedback ($Wald = .03$, $p = .87$), negative affect ($Wald = 1.52$, $p = .22$), disappointment score ($Wald = .20$, $p = .65$).)

backward elimination was used. Regret score turned out to be the only significant predictor, with higher regret scores being associated with an increased likelihood of deciding to take another card ($B = .45$, $Wald = 7.89$, $p < .01$).[3]

# DISCUSSION

Nearly half of the regret condition participants, correctly stopping at the score of 18 in the first round, switched their choice in the second round by deciding to take a further card, whereas most disappointment condition participants once more decided for the optimal alternative.

Our results therefore showed that choice switching can happen also following the unfortunate failure of an optimal choice (non-adaptive choice switching). Most important, choice switching turned out to be associated with the intensity of experienced regret, as showed in the regression analysis. Thus, we could expect an even stronger effect in real-life situations with a greater level of involvement.

# STUDY 2

## Method

### Participants and design

A total of 80 students (49 women and 31 men; mean age = 22.9 years, $SD = 2.69$) from Trieste University, who did not previously take part in Study 1, participated voluntarily. They were randomly assigned to three experimental conditions: control condition ($N = 25$), regret condition ($N = 28$) and disappointment condition ($N = 27$). Participants were tested individually and were advised about the data retained and that anonymity was fully ensured, no sensitive data were collected.

### Procedure

As in Study 1, participants played a first round of Blackjack with a 5-Euro prize and received a feedback (regret or disappointment) about the previous choice (to take another card or to stop at the score of 18). Afterwards, they played a round of Red & Black, a simple computerized card game similar to the one proposed originally by *Slovic (1966)* and used more recently in other studies (e.g., *Fernandez-Duque & Wifall, 2007*). In this game, participants were presented 10 cards lying face down into two rows on the computer screen, they were told there were nine red queens and a single black queen. Their task consisted in deciding how many and which cards to turn face up, by mouse-clicking on

the back of the cards, one by one: Every red queen increased their jackpot by 0.50 Euros, but the black queen (the "disaster" card) resulted in a total loss and ended the game. Participants could choose to finish playing and collect their jackpot at any moment by pressing the "stop" button. To obviate the possibility of participants finding the black queen too early on, the game was rigged to have the black queen appear only if participants decided to continue the game until the eighth card.

Participants in the control condition played only Red & Black, without the previous Blackjack round. Red & Black participants' behavior was diagnostic of the underlying regret mechanism. Differently from previous versions of this game used to investigate the effects of missed opportunities (e.g., *Büchel et al., 2011*; *Brassen et al., 2012*), our participants were not forced to turn the cards sequentially; instead, they were free to choose the cards in whatever order they preferred. Moreover, the position of the "disaster" card was not shown after participants stopped. This means that in our version of the game our participants should not feel regret for having missed the opportunity to gain more (e.g., counterfactuals like "I could have turned two more cards safely" are very unlikely). Thus, in this game the regret-minimizing behavior consists in stopping early, to avoid receiving the "disaster" card. If regret influences subsequent choices by increasing future regret aversion (indirect effect), participants in the regret condition should have engaged in regret minimizing behavior, by deciding to stop after turning over fewer cards than participants in the disappointment and control conditions. Alternatively, if regret influences subsequent choices by inhibiting a previously selected option (direct effect), regret condition participants were expected to avoid the decision causing regret in the previous Blackjack round (i.e., stopping), to the extent that that they would have turned over more cards than participants in the other conditions.

## Results

Most participants in the first round of Blackjack (50 out of 55, 91%) decided to stop after scoring 18. Thus, as in Study 1, participants correctly recognized this as the optimal decision. Data from the 5 participants who decided to continue and take another card were excluded from the analyses reported in the next section.

### Choices in Red & Black

Table 2 shows the mean and the median number of cards turned over in Red & Black for each of the three conditions. Data were analyzed using non-parametric tests, since regret condition's data were highly skewed. A Kruskal-Wallis test yielded a significant difference among the three conditions ($\chi^2_{(2,75)} = 7.83$, $p = .02$). Planned comparison revealed that participants in the regret condition turned over significantly more cards than participants in the disappointment condition (Mann–Whitney $Z = 2.30$, $p = .02$) and participants in the control condition (Mann–Whitney $Z = 2.53$, $p = .01$) did, with no significant difference, however, between disappointment and control condition participants (Mann–Whitney $Z = 0.41$, $p = .68$). Moreover, more than half of the participants in the regret condition (14 out of 25, 56%) lost the game by turning over cards until arriving at the "disaster" card (card nr. 8), whereas only 20% of the participants in

**Table 2** Mean and median of number of cards turned over in Red & Black in each of the three conditions.

|  | Mean | SD | Median |
|---|---|---|---|
| Control | 6.04 | 1.37 | 6 |
| Disappointment | 6.24 | 1.20 | 6 |
| Regret | 7.04 | 1.31 | 8 |

the other two conditions (5 out of 25 in both the control and disappointment conditions) continued until the eighth card.

## Discussion

Without a prior experience of regret (control and disappointment conditions), in the Red & Black game participants tended to stop after turning over approximately 6 cards; this behavior was not far from the game's actual optimal strategy (maximum expected value at 5 cards). Participants with a previous experience of regret (regret condition), however, decided to turn over more cards, and 56% of these lost the game by turning over eight cards.

This result has multiple implications. Firstly, a non-adaptive choice switching behavior induced by regret was once again confirmed. Secondly, this effect was not found to be domain-specific, thus regret experienced after a decision task can influence people's behavior in a different subsequent task (see also *Creyer & Ross, 1999*; *Raeva, Mittone & Schwarzbach, 2010*). Thirdly and most importantly, these findings support the hypothesis of a direct effect of experienced regret on subsequent choices: Regret leads to the rejection of the decision that previously resulted in a bad outcome. Thus, people are less likely to make this decision again in a subsequent choice, and this could happen regardless of its likelihood of success.

## CONCLUSIONS

Research has demonstrated that people try to avoid future experiences of regret by opting for behavior and choices that minimize the possibility of feeling this negative emotion (see, e.g., *Simonson, 1992*; *Beattie et al., 1994*). Yet, things do not always go as planned, and regret is unfortunately a common and somewhat painful experience that can influence future behavior: People are less likely to once more opt for a decision that previously led to regret, and will conversely select other alternatives. This behavior has been termed adaptive choice switching, since it is usually functional, allowing people to learn from their previous mistakes and decreasing the probability of repeating negative outcomes (*Baumeister et al., 2007*; *Zeelenberg & Pieters, 2007*; *O'Connor, McCormack & Feeney, 2014*). Yet, as demonstrated by the studies reported herein, after a regret experience people may avoid the previously made decision, even when it is still better than the other alternatives. In Study 1, participants played two rounds of Blackjack and in the second round almost half of them switched away from the good decision and selected the non-optimal decision as a consequence of having received a regret feedback at the end of the previous round. In Study 2, the same pattern emerged even when the second round of Blackjack was substituted with a different game, thus revealing a carry-over effect of experienced regret also on a

subsequent situation not directly related to the one that produced regret. Moreover, Study 2 shed light on the mechanism underlying this bias by using a game in which the two hypothesized mechanisms (direct effect vs. indirect effect) would have led to different behaviors. Results supported the hypothesis that regret influences subsequent choices by inhibiting the previous decision (direct effect): If this option becomes available again in a subsequent decision task, it is less likely to be chosen, regardless of its intrinsic value. We conversely found no evidence supporting the hypothesis that the effect of regret consists in priming the anticipation of regret in subsequent choices (indirect effect). This result is consistent with recent work conducted by *Raeva, Van Dijk & Zeelenberg (2011)*, who found no evidence of increased regret anticipation after a recent experience of regret, and by *O'Connor, McCormack & Feeney (2014)*, who found behavioral consequences of regret in children incapable of anticipating regret.

Overall, it appears that the experience of regret induces a strong tendency to avoid a decision that had previously led to regret, and that this can occur even when it remains objectively the best one available. Understanding the reasons people behave this way, abandoning a good decision for a non-optimal one after an experience of regret, is theoretically challenging. We argue that this effect is a bias resulting from the use of what we might call the "regret heuristic," a simple strategy that consists in basing actual choice on the outcome of the previous choice. This strategy could be viewed as a particular instance of the affect heuristic (*Finucane et al., 2000*), in which a previous decision is tagged with a negative affect after having learned, through explicit feedback (as in the present studies), or after having imagined (by counterfactual thinking), that a different decision would have been better. This regret-mediated information is highly available, thus the previous outcome can be used as a heuristic attribute in place of the target attribute, which in this instance is the option's likelihood of success (*Kahneman & Frederick, 2002*; *Kahneman, 2003*).

The findings presented in this paper highlight the possibility of the biased consequences of regret and suggest a new account for these effects under a broader perspective. We do not claim that this paper represents an exhaustive study, further research is obviously needed to explore the behavioral consequences of experienced regret and the consistency of the regret heuristic hypothesis.

### Funding
This work was carried out without any funding sources.

### Competing Interests
The authors declare there are no competing interests.

### Author Contributions
- Francesco Marcatto conceived and designed the experiments, performed the experiments, analyzed the data, contributed reagents/materials/analysis tools, wrote the paper, prepared figures and/or tables, reviewed drafts of the paper.

- Anna Cosulich conceived and designed the experiments, performed the experiments, contributed reagents/materials/analysis tools, reviewed drafts of the paper.
- Donatella Ferrante conceived and designed the experiments, analyzed the data, contributed reagents/materials/analysis tools, wrote the paper, reviewed drafts of the paper.

## Human Ethics

The following information was supplied relating to ethical approvals (i.e., approving body and any reference numbers):

Institutional review board approval for conducting empirical studies on human participants was not required by the institution. All participants were advised about the data retained and that anonymity was fully ensured. No sensitive data were collected.

## Supplemental Information

Supplemental information for this article can be found online at http://dx.doi.org/10.7717/peerj.1035#supplemental-information.

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
