# Peer review of "Once bitten, twice shy: experienced regret and non-adaptive choice switching"

_PeerJ, doi:10.7717/peerj.1035_

## Round 0.1 · original submission · Minor Revisions

Both reviewers expressed very favourable opinions of this work, with the bulk of the comments directed toward points in the Introduction and Discussion sections, which I believe could be addressed quite easily. In addition, both authors requested clarification of the definitions and analysis used as part of the regret measurements, and Reviewer 2 made several detailed suggestions, which I urge you to consider carefully in your revised manuscript. Although the level of revision suggested by the reviewers is relatively small, it is possible that changes in these definitions might affect the overall conclusions of the paper. Therefore, the amended manuscript will need to be re-reviewed before a final decision is made.

Reviewer 1 ·

Basic reporting

The introduction is very clear and well written.

The second paragraph in the regret and adaptive choices section does not seem to fit in with the paper as a whole as the authors don't discuss neruopsychological work in any further detail.

I think the authors could include a discussion about disappointment in the introduction section as their studies are assessing regret and disappointment but disappointment isn't discussed in any detail. I would question whether their definition of disappointment (i.e. you obtained a bad outcome and even the non-chosen alternative would have produced the same outcome) could also be interpreted as relief - i.e. you could not have done any better.

Experimental design

I feel the regret and disappointment scale could be described briefly, i.e. how many items are on the scale?

Are different participants used in Study 2?

It could be argued that disappointment may be felt in conjunction with regret in some instances. Further, the authors suggest that Blackjack may induce high disappointment. This is why I feel it is necessary to disentangle regret and disappointment in the introduction.

Validity of the findings

Good

Additional comments

A really nicely written paper.

If there is space I think the authors could reiterate that they are interested in experienced rather than anticipated regret.

Reviewer 2 ·

Basic reporting

The article is for the most part clearly written. The introduction is brief but concise and covers the main issues that need to be discussed. However, I have two comments on the introduction.
First, there is a very relevant paper that is not discussed: Brassen, Stefanie, et al. "Don’t look back in anger! Responsiveness to missed chances in successful and nonsuccessful aging." Science 336.6081 (2012): 612-614, plus a previous paper by the same group using the same task (i.e., a version of the Slovic task). Essentially in that study the authors look at something very similar to what is examined in the second experiment in the current paper, except they do this within the same task rather than switching between tasks. They find that participants who regret a previous missed chance are more likely to turn over more cards in the next trial (although due to the way the data are reported, it is hard to work out how large or robust an effect this is). Given the similarity of the Brassen et al. study to the current study, it is essential that the authors discuss it and make clear the distinctive contribution made by their own study.
Second, the authors argue in the introduction that their study will allow them to distinguish between direct and indirect effects of regret. I think that the authors need to preview in the introduction how they are going to examine this, as it is perhaps the more controversial claim in their paper.

Experimental design

The design of the experiment seems straightforward to me, although it would have been nice if the authors had taken regret ratings in the second experiment as well, as this would have strengthened their argument about the role of regret. I note also that the way the distinction between regret and disappointment is operationalized in this paper is somewhat different to how it is usually operationalized. In previous studies, regret is assumed to be based on a comparison between what one could have obtained if one had chosen differently, and what one did obtain; disappointment is assumed to be based on a comparison between what one might have hoped or expected to obtain and what one did obtain (e.g., Mellers et al., 1999). In these previous studies, participants typically have to choose between risky and safe gambles, and disappointment is taken to be the emotion that one feels if one gets the worse outcome from one's chosen gamble, whereas regret is the emotion that one feels if one sees one could have had a better outcome if one had chosen differently. In the case of disappointment, typically participants do not actually see what the counterfactual outcome would have been. By contrast, in this task participants do see the counterfactual in the disappointment condition. I'm not 100% sure why this is, and it would be good if the authors could make this clearer. This is an important issue, because the conclusions of the study in part depend on this comparison between conditions.

Validity of the findings

The analyses seem generally fine to me, although I would have liked to see more detail on the regression findings and some justification of doing backward stepwise regression.
What I found most difficult to grasp was the interpretation of the findings from Study 2. It is very interesting that there is a carry-over effect on risk-taking between different task types, and that is definitely worth reporting. However, the authors. interpretation hinges on the idea that those participants who are trying to avoid regret (i.e., the regretters from the previous task) will turn over the fewest cards; that fact that they instead turn over more cards is taken as evidence for a direct effect of regret. The problem with this interpretation, while interesting, is that participants might adjust their ideas regarding what behavior is likely to lead to regret, based on their experience in the first task. So, they might approach the second task by assuming that they are least likely to regret taking a risk. This would suggest that the effect is indirect rather than direct. I would have liked to see some more robust argument for the authors' interpretation.

---

## Round 0.2 · accepted · Accept

Thank you for your prompt and thorough responses to the reviewers' comments. I am pleased to accept your revised manuscript for publication.

Reviewer 2 ·

Basic reporting

This is satisfactory

Experimental design

This is satisfactory. There is a better explanation of the design in Study 2 now.

Validity of the findings

This is satisfactory

Additional comments

The authors have responded to my comments in a satisfactory way.